# Anomalous Properties of Cyclodextrins and Their Complexes in Aqueous Solutions

**DOI:** 10.3390/ma16062223

**Published:** 2023-03-10

**Authors:** Thorsteinn Loftsson, Hákon Hrafn Sigurdsson, Phatsawee Jansook

**Affiliations:** 1Faculty of Pharmaceutical Sciences, University of Iceland, Hofsvallagata 53, 107 Reykjavik, Iceland; 2Faculty of Pharmaceutical Sciences, Chulalongkorn University, 254 Phyathai Road, Pathumwan, Bangkok 10330, Thailand

**Keywords:** cyclodextrins, properties, aggregation, nanoparticles, microparticles, supersaturated solutions

## Abstract

Cyclodextrins (CDs) are cyclic oligosaccharides that emerged as industrial excipients in the early 1970s and are currently found in at least 130 marketed pharmaceutical products, in addition to numerous other consumer products. Although CDs have been the subject of close to 100,000 publications since their discovery, and although their structure and properties appear to be trivial, CDs are constantly surprising investigators by their unique physicochemical properties. In aqueous solutions, CDs are solubilizing complexing agents of poorly soluble drugs while they can also act as organic cosolvents like ethanol. CDs and their complexes self-assemble in aqueous solutions to form both nano- and microparticles. The nanoparticles have diameters that are well below the wavelength of visible light; thus, the solutions appear to be clear. However, the nanoparticles can result in erroneous conclusions and misinterpretations of experimental results. CDs can act as penetration enhancers, increasing drug permeation through lipophilic membranes, but they do so without affecting the membrane barrier. This review is an account of some of the unexpected results the authors have encountered during their studies of CDs as pharmaceutical excipients.

## 1. Introduction

About half a century ago, a group of cyclic oligosaccharides called cyclodextrins (CDs) emerged as enabling excipients for solubilization and stabilization of poorly soluble drugs in aqueous solutions. They rapidly gained the attention of pharmaceutical formulators in their constant search for new ways to solubilize and stabilize drug candidates. Currently, CDs can be found in at least 130 marketed pharmaceutical products, as well as in numerous other industrial products, including food, cosmetic, and toiletry products, most of which have been marketed during the past few decades [1,2,3,4,5]. CDs were first described as bacterial digest of starch by Antoine Villiers in 1891 [6]. In the following decades, their chemistry, structure, and physiochemical properties were investigated, including their ability to form water-soluble inclusion complexes of poorly soluble compounds [1,7,8,9,10,11,12,13]. In the beginning, only small amounts of relatively impure CDs were available, and this hampered their industrial applications; however, in the 1970s, biotechnological advances allowed large-scale production of pure CDs and their industrial application as enabling excipients. Since their discovery, CDs have been the subject of close to 100,000 publications, including about 61,000 peer-reviewed scientific articles and about 32,000 patents, most of which have been published during the past few decades (SciFinder; American Chemical Society, February 2023). Hence, scientific and industrial interest in CDs is significant and rapidly growing. Nevertheless, CDs are still surprising investigators with their unexpecting and, sometimes, hard-to-explain properties. This is an account of some of the unexpected results the authors have encountered during their studies of CDs as pharmaceutical excipients. It starts with a short theoretical background that might help to explain some of these CD oddities.

## 2. Cyclodextrins as Complexing Agents

The most abundant natural CDs are formed by 6 (αCD), 7 (βCD), and 8 (γCD) (1→4)-linked α-D-glucose units that form a doughnut-shaped oligosaccharide with a hydrophilic outer surface and a somewhat hydrophobic central cavity [14,15,16,17,18,19]. Natural CDs have rather limited solubility in water, but freely soluble CD derivatives have been synthesized that are better suited for, e.g., parenteral solutions (Table 1). In aqueous solutions, CDs form water-soluble inclusion complexes with many poorly soluble drugs (or other poorly soluble substrates) by taking up lipophilic moieties of the drug molecules into the hydrophobic cavity. No covalent bonds are broken or formed during the complex formation, and, in aqueous solutions, drug molecules bound within the complex are in dynamic equilibrium with unbound molecules where the complexes are constantly being formed and dissembled on a milli- to microsecond timescale [20]. The classification of drug/CD complexes is based on the work by Higuchi and Connors and their studies of solubilizing complexes and phase-solubility profiles (Figure 1) [21]. Water-soluble complexes form A-type phase solubility profiles. Linear A_L_-type phase-solubility profiles are observed when the complex formed is first-order with respect to the CD (i.e., the ligand) and first- or higher-order with respect to the drug (e.g., 1:1 and 2:1 drug/CD complexes). Profiles displaying positive deviation from linearity (i.e., A_P_-type profiles) are, for example, observed when the complex is first-order with respect to the drug and second- or higher-order with respect to the CD (e.g., 1:2 drug/CD complex). B-type phase solubility profiles are observed when the complex has limited solubility in water (Figure 1).

Most frequently, one drug molecule forms an inclusion complex with one CD molecule to give a 1:1 drug/CD complex (see figure in Table 1). In this case, a linear phase-solubility diagram is observed (i.e., A_L_-type) with a slope that is less than unity, and the stability constant (*K*_1:1_) can be obtained from the following equation:(1)$$K_{1:1\;}=\frac{Slope}{S_{0}\cdot \left(1\;-\;Slope\right)},$$
where *S*_0_ is the drug solubility in the aqueous solution when no CD is present, frequently referred to as the intrinsic drug solubility. *S*_0_ represents the thermodynamic equilibration solubility of the drug in the complexation medium in identical conditions (i.e., same pH, same temperature, etc.) when CD is not present. *S*_0_ is the y-intercept (i.e., *S_int_*) of the phase-solubility profile. Stepwise formation of 1:2 drug/CD complexes, where a 1:2 complex is formed following formation of a 1:1 complex, results in A_P_-type profiles where the stability constants (i.e., *K*_1:1_ and *K*_1:2_) can be obtained by, e.g., curve fitting to a quadratic model:(2)$$S_{tot}{=S}_{0\;}{+K}_{1:1}\cdot S_{0}\cdot \left[CD\right]{+K}_{1:1}\cdot K_{1:2}\cdot S_{0}\cdot {\left[CD\right]}^{2},$$
where [*CD*] represents the concentration of free CD molecules in the aqueous solution [21,23]. Higuchi and Connor did not work with CDs but based their classification system on studies of aqueous solubility of poorly soluble drugs, such as sulfathiazole, that form water-soluble but simple non-inclusion complexes with low-molecular-weight complexing agents, such as caffeine (MW 194 Da; logP −0.6) [24]. Such complexing agents have relatively few hydrogen bond acceptors (H-acceptors) and donors (H-donors), while the much larger CD molecules are very hydrophilic with numerous H-acceptors and H-donors (Table 1). Consequently, in aqueous solutions, CD molecules interact more strongly than the low-molecular-weight complexing agents, both with each other and with the surrounding water molecules.

## 3. The Anomalous Properties of Water

Water has some unique or anomalous physiochemical properties [25,26,27,28,29]. The water molecule consists of a highly electronegative oxygen atom that is covalently bound to two weakly electropositive hydrogen atoms that form a 104.5° angle, giving the molecule a close-to-tetrahedral structure. The electrostatic molecular surface forms a dipole where the oxygen atom is partially negative and the hydrogen atoms are partly positive. In liquid water, the polarity of each water molecule results in an intermolecular attraction and hydrogen bond formations. On average, each water molecule forms about 3.6 hydrogen bonds with surrounding water molecules; however, importantly, the exchange of hydrogen-bond partners via breaking and reforming occurs in a time range between 1 and 5 picoseconds (ps) [26,30]. Hydrogen bonds are relatively strong (~5–40 kJ/mol) compared to van der Waals interactions (~1–10 kJ/mol) but much weaker than covalent bonds (~200–1000 kJ/mol). Intermolecular hydrogen bonding of water molecules leads to enhanced molecular cohesion that affects the physiochemical properties. For example, the melting and boiling points of hydrogen sulfide (H_2_S; MW 34 Da) are −85 °C and −60 °C, respectively, compared to 0 °C and 100 °C, respectively, for water (H_2_O; MW 18 Da). In fact, if water did not possess this extensive molecular adhesion, its boiling point would not be 100 °C but about −90 °C. This extensive hydrogen bonding also affects other physiochemical properties of liquid water such as its dielectric constant (ε 78.5 at 25 °C), density (1.000 g/mL at 3.98 °C), surface tension, and heat of vaporization (40.65 kJ/mol), making them all higher than expected [25]. The dielectric constants of organic solvents, such as ethanol (ε 24.3 at 25 °C) and glycerol (ε 42.5 at 25 °C), are much lower than that of water. Due to self-ionization of the water molecule, protons and hydroxide ions diffuse through water at much faster rate than other ions. For example, protons in the form of hydronium ions (H_3_O^+^) diffuse about seven times faster through water than Na^+^, and OH^−^ diffuses about 2.5 times faster than Cl^−^ [25]. In solutions, water molecules form intermolecular and directional interactions that give rise to a wide variety of molecular networks that are constantly being formed and disassembled.

In aqueous solutions, hydration shells of structured water molecules are formed around solute molecules and at membrane surfaces [26,31]. In the case of hydrophilic solutes, such as polar molecules (e.g., CDs), the partly positive and partly negative surfaces of the water molecule form hydrogen bonds with the solute molecules. Formation of hydrogen bonds is thermodynamically favored. Nonpolar solutes, like many Class II drugs of the biopharmaceutics classification system (BCS), are hydrophobic and unable to form hydrogen bonds with water. Instead, water molecules form structured layers of water around the hydrophobic solute molecules that tend to aggregate to minimize the contact area of the hydrophobic surface with water. Hydration of hydrophobic solutes involves disruption hydrogen bonds and loss of entropy; thus, it is not thermodynamically favored, which can explain their low aqueous solubility. The structured water layers at membrane surfaces can hamper the permeation of drug molecules through membrane surfaces and decrease the rate of drug permeation through biomembranes such as mucosa [32,33,34,35].

Stable CD hydrates contain several water molecules. For example, in the solid state 2, 6, and 8.8 water molecules are present in the αCD, βCD, and γCD cavities, respectively, and 4.4, 3.6, and 5.4 water molecules, respectively, are spread around the CD exterior [22]. In bulk water, each water molecule forms on the average 3.6 hydrogen bonds, whereas, inside the αCD, βCD, and γCD cavities, each water molecule forms 1.5, 1.9, and 2.2 hydrogen bonds, respectively [36]. Thus, expulsion of water from the central CD cavity by a lipophilic drug moiety is thermodynamically favored. In aqueous solution, each glucose repeat unit of a CD molecule forms about 4–5 hydrogen bonds with surrounding water molecules, which create the first hydration shell, in addition to intramolecular hydrogen bonds between C2–OH on one glucose unit and C3–OH of the adjacent glucose unit [36,37]. It has been proposed that the ability of αCD, βCD, and γCD to form intra- and intermolecular hydrogen bonds can explain their differences in aqueous solubility (Table 1) [14,38,39]. Chemical substitution of OH groups of the CD molecules (e.g., in HPβCD and SBEβCD) has a significant effect on the hydrogen bonding of water molecules in the CD hydration shell [40]. Nevertheless, the main reason for the enhanced aqueous solubility of HPβCD and SBEβCD (Table 1) is the average degree of substitution (i.e., number of substituents per cyclodextrin molecule) and the random distribution of the substituents that results in amorphous mixtures of substituted CD isomers. The βCD molecule contains 21 hydroxyl functional groups that allow numerous possible combinations for substitution. Furthermore, the 2-hydroxypropyl moiety contains an additional optical center and, thus, the number of geometrical and optical isomers of HPβCD can be astronomical [41]. Studies have shown that both the molar substitution and the location of the substituents can influence the aqueous solubility and complexing behavior of HPβCD, as well as of other substituted CDs [42].

## 4. The Anomalous Properties of Cyclodextrins

CDs are small cyclic glucose polymers, or oligosaccharides, with molecular weight between 973 and 1297 Da. In spite of their rather trivial structure, their physiochemical properties have been shown to be far from predictable. In fact, these rather simple molecules are constantly surprising pharmaceutical formulators with their unexpected properties, especially in aqueous solutions, most of which can be explained by their inter- and intramolecular hydrogen bonding, as well as by the anomalous properties of water. It should also be mentioned that, while theoretical chemistry is frequently based on studies in close-to-ideal solutions (i.e., very dilute solutions), pharmaceutical solutions are nonideal solutions (i.e., concentrated solutions of drugs and excipients) and, thus, can deviate from the classical physicochemical principles.

### 4.1. Aggregation of CDs and CD Complexes

Like other saccharides and carbohydrate polymers, CD molecules and their complexes self-associate in aqueous solutions to form nanoparticles, even at relatively low CD concentrations [43,44,45,46,47,48]. Although association of CD molecules was first mentioned in the scientific literature some 40 years ago [49], and although formation of nanosized CD aggregates is observed at CD concentrations below 1% (Table 2), the effects of CD aggregates and CD complex aggregates are frequently overlooked. The reason might be that the aggregate weight fraction of the total CD mass can be low (e.g., 0.001%), and their diameters are frequently well below the wavelength of the visible light; thus, CD solutions appear clear even though they contain CD aggregates of various types and shapes [45,50,51,52]. Furthermore, nanosized aggregates are frequently transient and, thus, can be difficult to detect using conventional analytical techniques [48]. Depending on their size and physicochemical properties, the observed aggregates can be characterized as unstable nanoclusters of CD molecules and CD complexes that are constantly being formed and dissembled in aqueous solutions [43,44], as somewhat stable nano- or microparticles [53], and as unstable macroclusters [54] (Table 3). In most cases, the formation of nanosized CD complex aggregates does not affect the shape of the phase-solubility diagrams. However, due to CD complex aggregation, A_L_-type phase-solubility diagrams with slope less than unity do not necessarily indicate formation of 1:1 drug/CD inclusion complexes, and A_P_-type diagrams do not necessarily indicate formation of 1:2 drug/CD inclusion complexes [48,55,56]. Drug/CD complexes of more than one stoichiometry can also coexist in aqueous solutions [57].

The aggregation and hydration of CDs in aqueous solutions can also affect the rate of drug dissolution in aqueous CD media by decreasing the ability of dissolved CD molecules to form water-soluble inclusion complexes. Both CD aggregation and CD hydration decrease with increasing temperature, while the intrinsic solubility of both the drug and CD increases. Thus, heating the aqueous complexation medium, followed by equilibration at the desired temperature, can result in more reliable solubility data and better solubilization [23,58].

**Table 2 materials-16-02223-t002:** The apparent critical aggregation constant (cac) of some CDs at ambient temperature in pure water as determined by permeation through a semipermeable membrane with MWCO of 3.3–5 kDa [44], proton nuclear magnetic resonance (^1^H-NMR) [59], and dynamic light scattering (DLS) [59].

Cyclodextrin	Critical Aggregation Concentration (% *w*/*v*)		
	Permeation	^1^H-NMR	DLS
αCD	2.5		
βCD	0.8		
γCD	0.9		
HPβCD	11.8	2.1	1.8
SBEβCD		1.9	2.5
HPγCD		2.2	1.7

**Table 3 materials-16-02223-t003:** Classification of aggregates that have been observed in aqueous CD solutions. Adapted with permission from Ref. [44].

Type of Aggregate	Approx. Diameter (µm)	Properties	When Observed
Nanoclusters	<0.5	Transient clusters that are unstable and dissemble upon agitation and filtration.	Aqueous CD solutions. The nanoclusters consist of free CD and/or drug/CD complexes.
Nanoparticles	<0.5	Somewhat stable particles that can tolerate centrifugation and filtration but dissemble upon medium dilution.	Aqueous αCD, βCD, and γCD solutions, frequently at relatively low CD concentrations. The nanoparticles consist of free CD and/or drug/CD complexes.
Microparticles	1–50	Stable particles that can precipitate from aqueous media but disassemble upon medium dilution.	Aqueous αCD, βCD, and γCD solutions, containing poorly soluble drugs and at relatively high CD concentrations. The microparticles consist of drug/CD complexes.
Macroclusters	>1000	Very unstable transient clusters that can be visible to the naked eye but dissemble upon agitation and filtration.	Transient clusters (or transient particulate matter) that are sometimes observed in aqueous parenteral solutions containing a relatively high concentration of water-soluble CD derivatives, such as HPβCD.

### 4.2. S_0_ Is Not Always Equal to S_int_

One of the basic assumptions of the phase-solubility technique used to determine the stability constants of solubilizing complexes is that the solubility of a drug (i.e., the substrate) in the aqueous complexation medium (i.e., the intrinsic solubility or *S*_0_) when no complexing agent (i.e., ligand) is present is equal to the y-intercept (*S_int_*) of the phase-solubility diagram (Figure 2A) [21]. However, this is not always the case, especially for lipophilic drugs with limited solubility in water. Such molecules tend to form small aggregates in aqueous solutions. Even many hydrophilic drugs of poor aqueous solubility (e.g., doxorubicin, solubility 0.5 mg/mL and logD −1.6 at pH 7) form aggregates in aqueous solutions [60]. In aqueous solutions, most often only the drug monomer can enter the CD central cavity to form an inclusion complex (Figure 2B), while the observed intrinsic drug solubility (*S*_0_) represents the concentration of both drug monomers and nanosized drug aggregates; thus, *S*_0_ can be greater than *S_int_* (Figure 2C). Consequently, values of stability constants of drug/CD complexes that are derived from phase-solubility diagrams are frequently incorrect, such as values of *K*_1:1_ obtained from Equation (1). The complexation efficiency (*CE* in Figure 2C) of a given CD at a given condition only depends on the slope of the phase-solubility diagram and, thus, is independent of both *S*_0_ and *S_int_* [61]. In addition to drug aggregation, drug–excipient interactions (e.g., formation of water-soluble drug–polymer complexes) can result in *S*_0_ > *S_int_* [62,63].

### 4.3. The Value of K Is Method-Dependent

Both the stoichiometry and the aggregation of drug/CD complexes are affected by the drug and CD concentrations in the aqueous complexation medium. In addition, analytical methods applied to determine the stability constants (i.e., the *K* values) can be affected by the CD and drug/CD complex aggregation. Thus, experimental *K* values can be both concentration- and method-dependent [64,65,66]. For example, the phase-solubility diagrams of diflunisal in pure aqueous HPβCD solutions have slopes between 1.2 and 1.3, indicating 2:1 stoichiometry (i.e., [diflunisal]:[HPβCD] ratio) of the diflunisal/HPβCD complex, although other studies have clearly shown that only 1:1 inclusion complexes are formed in the solutions [56]. A possible explanation can be that additional diflunisal solubilization is obtained through non-inclusion complexation and/or a micellar-like solubilization by diflunisal/HPβCD complex aggregates. It is known that diflunisal/HPβCD complexes aggregate in aqueous solutions [55,56]. The value of *K*_1:1_ for the diflunisal/HPβCD complex can be obtained by other techniques but the value is highly method-dependent (Table 4).

### 4.4. Water-Soluble Polymers Can Enhance the Solubilizing Efficiency of CDs

Polymers are known to interact with a variety of particulate systems such as nanoparticles, micelles, and cyclodextrin aggregates [69,70,71]. It is also well documented that polymers enhance micellar solubilization of lipophilic compounds in aqueous media [72,73,74]. Some years ago, it was observed that, in aqueous solutions, polymers have synergistic effects on CD solubilization of poorly soluble drugs, and it is assumed that this effect is analogous to that of micellar solubilization, and that the effect is related to the aggregation of CD complexes [75,76,77,78,79]. To avoid the formation of polymer–CD inclusion complexes and rotaxanes that might result in decreased *CE*, the diameter of the polymer should preferably be greater than that of the CD cavity [80]. Then, through hydrogen bonding, the polymers stabilize the drug/CD complex aggregates, as well as individual drug/CD complexes, resulting in enhanced complexation efficacy and increased drug solubilization [77]. If the drug/CD complex has limited solubility in aqueous complexation media, addition of a water-soluble polymer will enhance the solubility through solubilization of the nanosized drug/CD complex aggregates [81]. For example, the aqueous solubility of βCD in pure water was determined to be 18.6 ± 0.4 mg/mL when no PVP was present, increasing to 20.5 ± 0.8 mg/mL when 2.5 mg/mL PVP was present (~10%) [81]. However, when the aqueous medium is saturated with both βCD and carbamazepine, the βCD solubility is increased by about 150% (Figure 3). The figure shows that the solubility of βCD is about 18 mg/mL in pure water, about 33 mg/mL when the solution is saturated with both βCD and carbamazepine, and about 53 mg/mL when 0.75 mg/mL of PVP is added to the βCD and carbamazepine-saturated solution. Furthermore, the solubility of carbamazepine in the saturated βCD solution increases from about 2.2 to 4.6 mg/mL when 1 mg/mL PVP is present.

### 4.5. Effects of Ions on βCD Solubilization of Drugs

Hydroxy acids, such as lactic acid, citric acid, and tartaric acid, enhance the aqueous solubility of βCD due to their ability to influence intra- and intermolecular hydrogen-bonding [82]. Studies have shown that hydroxy acids enhance the complexation efficacy and aqueous solubility of the complex when a poorly soluble basic drug forms a salt with the acid while the acid is able to interact with the hydroxyl groups on the wider external rim of a CD molecule [83]. However, other organic acids and bases, such as sodium acetate, sodium benzoate, and benzalkonium chloride, are also able to enhance the solubility of drug/βCD complexes, presumably through solubilization of drug/βCD aggregates, even for drugs that are unable to form salts [84]. For example, the aqueous solubility of hydrocortisone (a neutral drug) at room temperature was determined to be 0.4 mg/mL in pure water, 2.1 mg/mL in aqueous 4% (*w*/*v*) βCD solution, and 7.1 mg/mL in aqueous 4% (*w*/*v*) βCD solution containing 1% (*w*/*v*) sodium acetate (Figure 4).

### 4.6. In Aqueous Solutions, CDs Frequently Resemble Organic Cosolvents

In chemistry, the word *complex* usually refers to molecules or molecular assembles formed by association of substrates (e.g., a drug molecule) and ligands (e.g., a CD molecule); in aqueous solutions, noncovalent complexes are in dynamic equilibrium with unbound substrates and ligands [85]. With only few exceptions, drug/CD complexes have stability constants (e.g., *K*_1:1_) below 10^4^ M^−1^, and the complexes are transient in aqueous solutions, constantly being formed and dissembled on a milli- to microsecond timescale [20]. Consequently, the release of a drug molecule from the CD cavity will generally not be a limiting factor, such as during drug permeation from an aqueous CD solution through a lipophilic biological membrane. One exception is the effect of CDs on the thermodynamic activity of drugs in aqueous solutions, which can result in reduced drug permeation [86]. In this respect, the effect of CDs on drug permeation from the aqueous exterior through biological membranes is similar to that of organic cosolvents. It is common practice in reversed-phase high-performance liquid chromatography to increase the concentration of organic solvents, such as methanol and acetonitrile, in the aqueous mobile phase to shorten the drug retention time. The same effect can be obtained by adding CDs, such as βCD and HPβCD, to the mobile phase [66,87]. Numerous studies have shown that the transient drug/CD complexes with *K*_1:1_ below 10^5^ M^−1^ do not affect the drug pharmacokinetic parameters after parenteral administration such as the half-life and volume of distribution [54,88]. In aqueous parenteral solutions, CDs behave as organic cosolvents. This is due to the very rapid release of drug molecules from the CD cavities upon intravenous administration, as well as the consequent medium dilution and drug binding to plasma proteins [89]. Although one would expect that aqueous drug/CD complex solutions behave like disperse systems (e.g., colloids), these and other observations indicate that, due to the transient nature of CD complexes and their aggregates, aqueous CD solutions can resemble aqueous solutions containing organic cosolvents.

### 4.7. CDs as Permeation Enhancers

Biomembranes, such as mucosa, form membrane barriers toward drug permeation from, e.g., the gastrointestinal tract to the systemic blood circulation. Although drugs can be transported actively through biomembranes, the capacity of the active absorption is frequently limited; thus, passive drug permeation through the membrane is dominating. Most often, membrane barriers consist of an unstirred water layer (UWL) at the exterior surface of a lipophilic membrane [33]. In the gastrointestinal tract, the UWL is the relatively thick (can be several hundred µm in humans) and viscous mucus layer that covers the lipophilic mucosal epithelium [90,91]. Figure 5 is a schematic presentation of permeation of a lipophilic drug through a simple two-layer biomembrane showing passive drug permeation first through an UWL and then through a lipophilic membrane. Passive drug permeation follows Fick’s first law that states that the drug flux (*J*) is the product of the permeation coefficient (*P*) and drug concentration gradient over the barrier. A greater concentration gradient results in faster permeation of the drug molecules (i.e., a larger value of *J*) [86]. A higher concentration of a dissolved lipophilic drug in the aqueous gastrointestinal fluid results in faster absorption of the drug through the mucosa into the systemic blood circulation and more complete absorption (i.e., higher drug bioavailability). In vitro studies have shown that CDs only enhance drug permeation through artificial membranes (e.g., PAMPA) and cellular tissues (e.g., Caco-2 membrane) when the permeation resistance of the UWL at the membrane surface contributes significantly to the overall permeation resistance of the membrane [92,93,94]. Moreover, the dissolution rate also affects the overall bioavailability, and CDs usually enhance the drug dissolution rate. An understanding of absorption barriers of a given drug is, however, essential for optimal formulation design [95]. The diffusional and kinetic parameters for drug diffusion through the UWL from aqueous CD formulations can be predicted by a simple experimental/mathematical approach [96]. In general, CDs enhance the permeation without affecting the barrier function of the mucosa. Most permeation enhancers increase the delivery of drugs through mucosal membranes by permeating into the lipophilic membrane where they disrupt the membrane structure, which results in faster drug permeation. CDs are very hydrophilic and, thus, unable to partition from the aqueous exterior into the lipophilic membrane. Furthermore, maximum absorption is obtained when the aqueous exterior is saturated with the drug, i.e., when *C_A_* is equal to drug-saturated solubility (Figure 5) [86]. Under such conditions, the dissolved drug molecules have the maximum ability to leave the aqueous exterior and partition into the lipophilic membrane. Lastly, it has been noted that drug/CD complex aggregates can reduce drug irritation in the gastrointestinal tract [97].

### 4.8. CDs Can Stabilize Supersaturated Solutions

When the concentration of dissolved drug in an aqueous solution is higher than the drug’s thermodynamic equilibrium solubility, the solution is said to be supersaturated. A supersaturated drug solution is unstable and can result in rapid drug precipitation. Pharmaceutical excipients that inhibit nucleation and crystal growth are referred to as drug precipitation inhibitors and include numerous polymers, such as hydroxypropyl methylcellulose and polyvinylpyrrolidone, cocrystals, and surfactants, as well as CDs [98,99,100]. The mechanism of inhabitation may include changes in surface tension, an increase in viscosity, and adsorption to the crystal layer, thereby blocking crystal growth. Although several common pharmaceutical excipients can stabilize supersaturated solutions, CDs have been shown to both enhance the drug dissolution rate and stabilize the resulting supersaturated drug solution.

According to the Noys–Whitney equation, the dissolution rate (*dC*/*dt*) is proportional to the difference between the drug concentration in the aqueous solution (*C_t_*) at time t and the drug solubility at the solid drug surface (*S*) [101].
(3)$$\frac{dC}{dt}=k\cdot (S\;-\;C_{t}),$$
where *k* is a constant. *dC*/*dt* increases with increasing *S* and decreases with increasing *C_t_*. *S* of poorly soluble drugs can, for example, be increased through the formation of salts of ionizable drugs, by converting crystalline drugs to amorphous ones or through the addition of surface-active agents to the solid drug dosage form, but also through formation of water-soluble drug/CD complexes. In some cases, the drug dissolution is fast enough to produce drug concentrations that are above the thermodynamic equilibrium solubility (*S*_0_). In this case, *C_A_* is greater than *S*_0_, resulting in even faster drug permeation (Figure 5). Figure 6 shows how both HPβCD and SBEβCD form supersaturated itraconazole solutions that are stable for at least 120 min. TPGS forms a supersaturated solution that is stable for about 40 min, while the other surface-active agents tested form instable supersaturated itraconazole solutions. Other investigators have observed that drug/CD complexes both enhance the rate of drug dissolution and stabilize the supersaturated drug solution formed from about 0.5 to about 12 h [102,103,104]. The influence of CDs on drug dissolution has been referred to as the spring (i.e., fast dissolution) and parachute (i.e., stabilization of supersaturated drug solution) effect [105,106].

## 5. Conclusions

Studies have shown that CDs possess peculiar properties that make them rather unique pharmaceutical excipients. CDs are solubilizing complexing agents that increase the aqueous solubility of poorly soluble drugs while also behaving like organic cosolvents such as ethanol. CDs and their complexes can self-assemble to form unstable aggregates that are able to solubilize poorly soluble compounds in a micellar-like fashion. Nanosized CD aggregates can self-assemble further to form stable microparticles that allow for sustained drug delivery after topical administration. CDs can enhance drug delivery through biological membranes without affecting the membrane barriers. CDs can increase drug dissolution in aqueous media and stabilize super-saturated solutions. Few if any other pharmaceutical excipients possess such diverse properties desired by drug formulators. However, at the same time, these properties can result in erroneous interpretations of experimental results.

## Figures and Tables

**Figure 1 materials-16-02223-f001:**
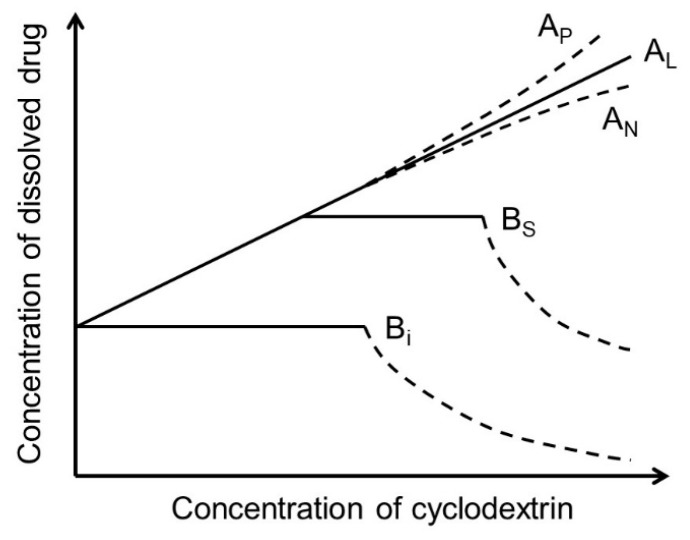
Phase-solubility profiles and classification of drug/CD complexes according to Higuchi and Connors [21].

**Figure 2 materials-16-02223-f002:**
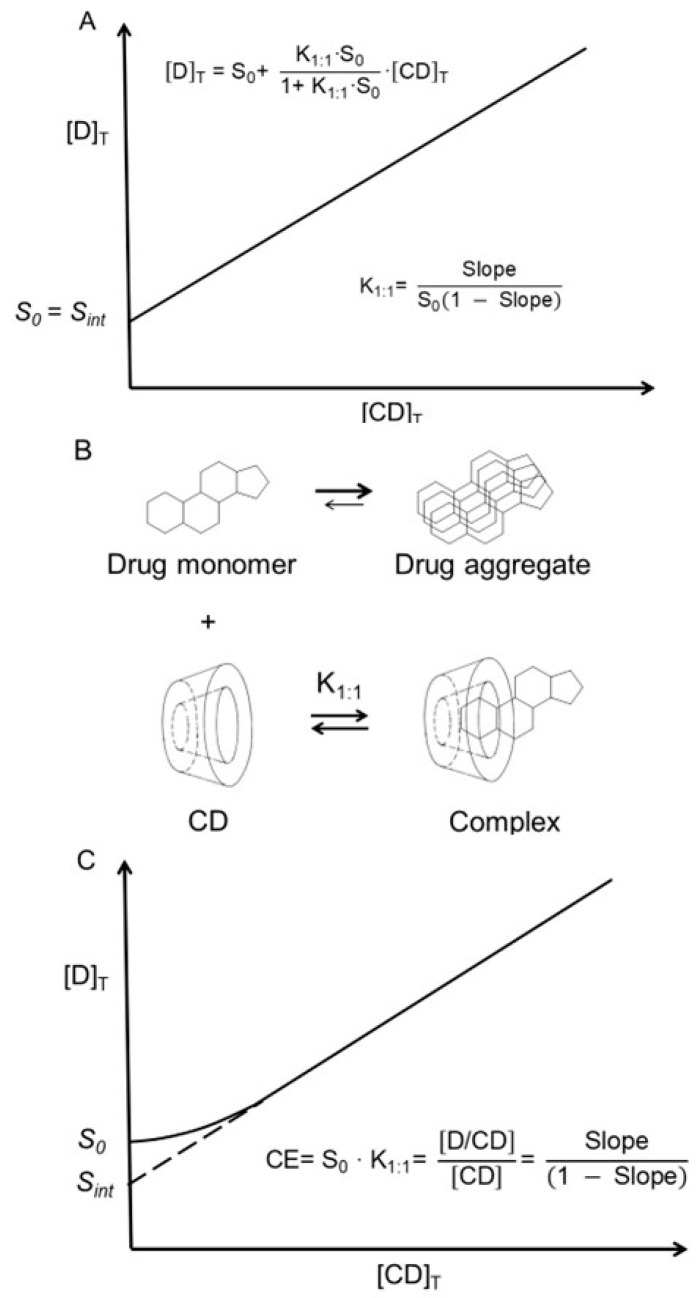
In general, one drug molecule (D) forms a complex with one CD molecule in pure water, and, if the complex is water-soluble, an A_L_-type phase-solubility diagram is observed (Table 1). (**A**) Classical A_L_-type phase-solubility profile of a 1:1 drug/CD complex (D/CD) according to Higuchi and Connors [21]. [D]_T_ is the total concentration of dissolved drug, and [CD]_T_ is the total concentration of CD in solution. In this case, S_0_ is equal to the intercept (*S_int_*). (**B**) Aggregation of drug molecules in water where only the monomeric drug can form a 1:1 complex with CD. (**C**) Drug aggregation, where *S*_0_ is the concentration of not only dissolved monomeric drug but also dissolved nanosized drug aggregates (*S*_0_ > *S_int_*). Here, the stability constant (*K*_1:1_) of the complex cannot be determined from the slope and intercept, and the complexation efficiency (*CE*) is used to evaluate the CD solubilization [61].

**Figure 3 materials-16-02223-f003:**
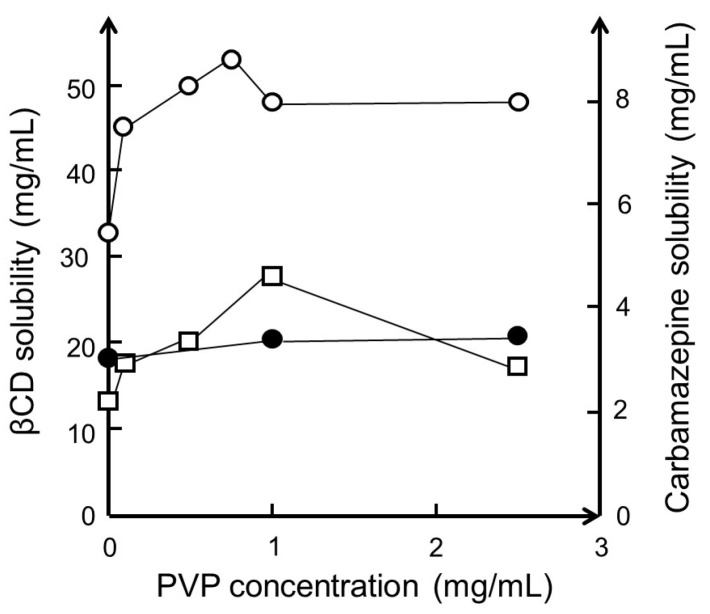
The effect of polyvinylpyrrolidone with a molecular weight of 40,000 (PVP) on the solubility of βCD and carbamazepine in water at room temperature (23 °C). Solubility of βCD when no carbamazepine is present (●). Solubility of βCD (○) and carbamazepine (□) when the complexation medium contains both excess βCD and excess carbamazepine. Unpublished results based on experiments described in [81].

**Figure 4 materials-16-02223-f004:**
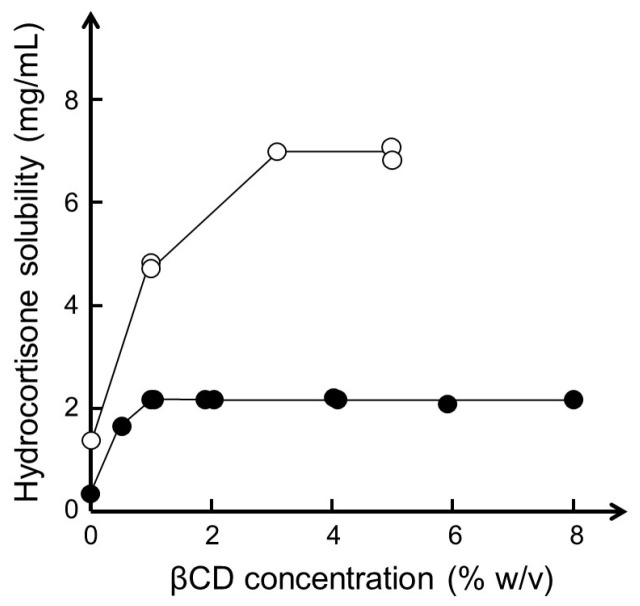
Phase-solubility diagram of hydrocortisone in aqueous βCD medium with (○) and without (●) 1% (*w*/*v*) sodium acetate at room temperature (22–23 °C) and pH 6.9. Based on data with permission from [84].

**Figure 5 materials-16-02223-f005:**
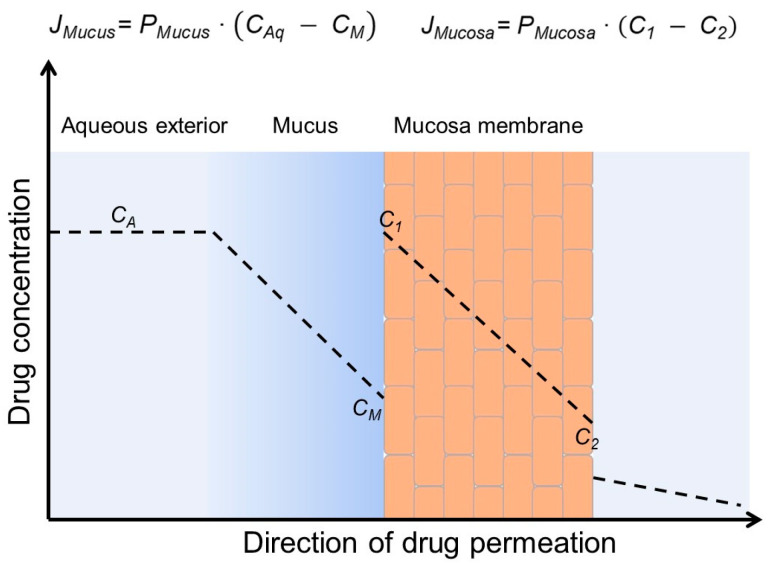
Passive permeation of a lipophilic drug from an aqueous exterior through mucus (a viscous unstirred water layer) to the mucosal surface where the drug partitions into the mucosa (a lipophilic membrane barrier) and then permeates the mucosa. The concentration gradient within the mucus layer (*C_A_* − *C_M_*) is the driving force for drug permeation through the mucus layer (i.e., diffusion barrier), and the concentration gradient within the mucosa (*C*_1_ − *C*_2_) is the driving force for drug permeation through the mucosa (i.e., membrane barrier). *J_Mucus_*, *J_Mucosa_*, *P_Mucus_* and *P_Mucosa_* are the drug fluxes and permeation coefficients through the aqueous mucus layer and the mucosa, respectively.

**Figure 6 materials-16-02223-f006:**
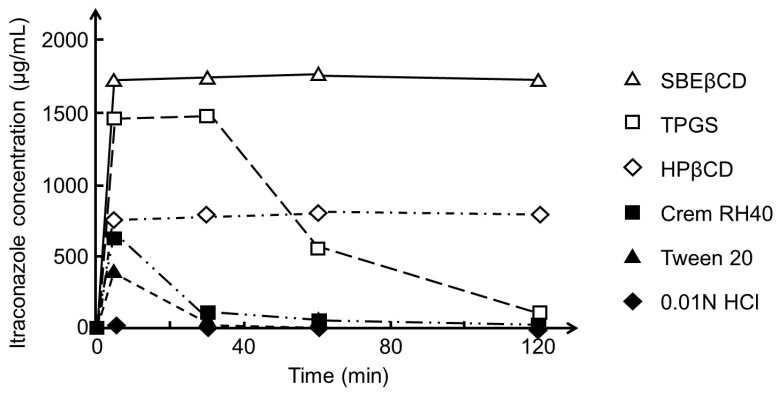
Effect of sulfobutylether β-cyclodextrin (SBEβCD), α-tocopheryl succinate esterified to polyethylene glycol 1000 (TPGS), 2-hydroxypropyl-β-cyclodextrin (HPβCD), Cremophor RH40 (Crem RH40), and polysorbate 20 (Tween 20) in aqueous 0.01 M HCl solution on the solubility of itraconazole. An excess amount of dissolve itraconazole (50 mg/mL) in dimethylformamide (DMF) was added to a medium containing no or 2.5% (*w*/*v*) of the excipients, and the itraconazole concentration was monitored for 120 min. The equilibrium solubility (*S*_0_) of itraconazole was determined to be 21 µg/mL in the HPβCD medium and 206 µg/mL in the SBEβCD medium. Adapted from Brewster et al. with permission [107].

**Table 1 materials-16-02223-t001:**
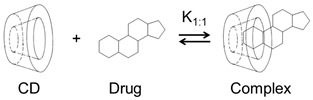
Cyclodextrins with monographs in USP–NF and Ph. Eur., and formation of 1:1 drug/CD inclusion complex.

Cyclodextrin	Abbrev.	Pharmacopoeia Name	Molecular Weight ^b^	Solubility in Water (mg/mL) ^e^	LogP ^f^	H-Acceptors ^g^	H-Donors ^g^
α-Cyclodextrin	αCD	Alfadex	973	129.5	−13	30	18
β-Cyclodextrin	βCD	Betadex	1135	18.4	−14	35	21
2-Hydroxypropyl-βCD	HPβCD	Hydroxypropylbetadex	1400 ^c^	>600	−11	39 ^c^	21 ^c^
Sulfobutyl ether βCD sodium salt	SBEβCD	Sulfobutylbetadex sodium	2163 ^d^	>500	<−10	53 ^d^	15 ^d^
γ-Cyclodextrin	γCD	Gammadex ^a^	1297	249.2	−17	40	24

^a^ USP43–NF38’s official name is “gamma cyclodextrin”. ^b^ Molecular weight of anhydrous cyclodextrin. ^c^ HPβCD with an average degree of substitution of 4.5. ^d^ SBEβCD with an average degree of substitution of 6.5. ^e^ Solubility in pure water at 25 °C [22]. ^f^ Calculated log_10_ of the n-octanol/water partition coefficient (SciFinder (scifinder.cas.org); American Chemical Society, November 2022). ^g^ Hydrogen bond acceptors (H-acceptors) and hydrogen bond donors (H-donors).

**Table 4 materials-16-02223-t004:** The stability constant (*K*_1:1_) of diflunisal/HPβCD complex at 23 to 25 °C and neutral pH (i.e., the pH ranged from 7 to 7.4). At 25 °C, the pKa of diflunisal is 2.9 and, thus, the drug is fully ionized at neutral pH.

Method	*K*_1:1_ (M^−1^)	Ref.
^19^F-NMR	2034 ± 403	[65]
Equilibrium dialysis	3892 ± 360	[67]
Potentiometry	5570 ± 40	[68]
Microcalorimetry	3394	[67]
UV spectroscopy	5069	[65]

## Data Availability

Not applicable.

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
