# Peer review of "Anomalous Properties of Cyclodextrins and Their Complexes in Aqueous Solutions"

_materials, 2023, doi:10.3390/ma16062223_

Round 1

Reviewer 1 Report

The review article is focused on the aggregation of cyclodextrins in water and the consequences of this aggregation. The manuscipt is very well written and brings comprehensive information on the behavior of cyclodextrins in aqueous solutions to scientists who are aware of this issue. I found only a few inaccuracies in the manuscript:

Table 1, lines 76 and 77. Calculate degree of substitution (DS)(i.e. number of substituents per cyclodextrin) and use it instead of "molar substitution".

Lines 154-156 - It should be stated, that the numbers are for solid state and not for a solution.

Chapter 4 (line 178 and further) - There is a different spacing than in the previous chapters.

Line 169 - Please use degree of substitution (see above). The degree of substitution is much more widely used in the literature, than the reported molar substitution.

Line 172 - It is not true. The authors did not take symmetry into account. E.g. There are only 3 possible monosubstituted isomers (and not 7 times more (21; i.e. the same for each glucose unit). Please correct the number.

Line 176 - It is not the only case of HPBCD, but also of other substituted cyclodextrins. There are reports for e.g. monosubstituted carboxymethylated cyclodextrins, how associations constants depend on position of substituent and the cavity size (type of CD) as well. Due to these differences, it is necessary to pay attention to individual batches during applications, as they can vary greatly in their properties (this has already been published as well). Please change the paragraph accordingly.

Line 180 - Authors could use exact numbers (they are using the exact numbers in the Table 1 anyway)

Line 443 - use "M" instead of "N". Normal solutions are obsolete and authors should use molar numbers instead.

Summary

Based on the above, I recommend the manuscript to be accepted for publication after minor corrections."

Author Response

Thank you.  Following is my response.

Table 1, lines 76 and 77. Calculate degree of substitution (DS)(i.e. number of substituents per cyclodextrin) and use it instead of "molar substitution".
Done.

Lines 154-156 - It should be stated, that the numbers are for solid state and not for a solution.
We have added “in the solid state” in line 154.

Chapter 4 (line 178 and further) - There is a different spacing than in the previous chapters.
Yes, and I do not know how to correct is!

Line 169 - Please use degree of substitution (see above). The degree of substitution is much more widely used in the literature, than the reported molar substitution.

Done.  It now reads “… is the average degree of substitution (i.e., number of substituents per cyclodextrin molecule) …”.

Line 172 - It is not true. The authors did not take symmetry into account. E.g. There are only 3 possible monosubstituted isomers (and not 7 times more (21; i.e. the same for each glucose unit). Please correct the number.

Done.  Numerical values have been removed.  It now reads “… allow numerous possible combinations for substitution”.

Line 176 - It is not the only case of HPBCD, but also of other substituted cyclodextrins. There are reports for e.g. monosubstituted carboxymethylated cyclodextrins, how associations constants depend on position of substituent and the cavity size (type of CD) as well. Due to these differences, it is necessary to pay attention to individual batches during applications, as they can vary greatly in their properties (this has already been published as well). Please change the paragraph accordingly.

You are correct.  We have added “… HPβCD as well as of other substituted CDs [42].”

Line 180 - Authors could use exact numbers (they are using the exact numbers in the Table 1 anyway)
Done.

Line 443 - use "M" instead of "N". Normal solutions are obsolete and authors should use molar numbers instead.

Done.

Reviewer 2 Report

Thorsteinn Loftsson et al. submitted an interesting review upon the CD and CD complex in water solution. The topic was interesting, and might be considered for publication in Materials, after Major Revisions. My comments:

1-      There were 9 keywords. In general, 5~6 keywords would be acceptable. Please consider to reduce the keywords.

2-      It was suggested to showcase the molecular structure of CDs in the Introduction Section.

3-      Nowadays, there were many other host-guest systems. Please make a proper comparison between CDs and others, in the appropriate place of the text.

4-      Please demonstrate the source of information in Table 1.

5-      For different classifications of CD aggregates, it was advisable to draw schematic illustrations about their structures.

6-      Figure 3 seemed to contain unpublished results. Please check the journal’s policy to see whether unpublished results were allowed in Reviews.

7-      Why did the lower two lines not contain SD values in Table 4?

8-      Please add your personal perspectives in the Conclusion Section.

Author Response

Thank you.  Following is my response.

1-      There were 9 keywords. In general, 5~6 keywords would be acceptable. Please consider to reduce the keywords.

Done.

2-      It was suggested to showcase the molecular structure of CDs in the Introduction Section.

The authors do not think it is needed as this manuscript is intended for publication in a special issue of the journal on cyclodextrins.

3-      Nowadays, there were many other host-guest systems. Please make a proper comparison between CDs and others, in the appropriate place of the text.

     We think that other types of inclusion complexes (e.g., calixarenes) are beyond the scope of this review on some anomalous properties of cyclodextrins.  For example, we have not observed calixarene aggregates.

4-      Please demonstrate the source of information in Table 1.

     This is clearly stated in the notes below the table. H-donors, H-acceptors, and MW are just calculated values.

5-      For different classifications of CD aggregates, it was advisable to draw schematic illustrations about their structures.

     The structure is unknown.  The classification is based on their properties, not their structure.

6-      Figure 3 seemed to contain unpublished results. Please check the journal’s policy to see whether unpublished results were allowed in Reviews.

     Figure 3 is original.  The data has not been published before. We are only referring to the method we used to obtain these unpublished data.

7-      Why did the lower two lines not contain SD values in Table 4?

     The original data do not contain SD values.

8-      Please add your personal perspectives in the Conclusion Section.

We think that our personal perspectives are already included in, for example, the last two sentences of the section: “Few if any other pharmaceutical excipients possess such diverse properties desired by drug formulators.  However, at the same time these properties can allow erroneous interpretations of experimental results.”

Round 2

Reviewer 2 Report

I have no further suggestions.